# miQC: An adaptive probabilistic framework for quality control of single-cell RNA-sequencing data

Ariel A. Hippen[1], Matias M. Falco[2], Lukas M. Weber[3], Erdogan Pekcan Erkan[2], Kaiyang Zhang[2], Jennifer Anne Doherty[4], Anna Vähärautio[2]*, Casey S. Greene[5], Stephanie C. Hicks[3]*

**1** Department of Systems Pharmacology and Translational Therapeutics, Perelman School of Medicine, University of Pennsylvania, Philadelphia, Pennsylvania, United States of America, **2** Research Program in Systems Oncology, Research Programs Unit, Faculty of Medicine, University of Helsinki, Helsinki, Finland, **3** Department of Biostatistics, Johns Hopkins Bloomberg School of Public Health, Baltimore, Maryland, United States of America, **4** Huntsman Cancer Institute and Department of Population Health Sciences, University of Utah, Salt Lake City, Utah, United States of America, **5** Department of Biochemistry and Molecular Genetics, University of Colorado School of Medicine, Aurora, Colorado, United States of America

* anna.vaharautio@helsinki.fi (AV); shicks19@jhu.edu (SCH)

**Data Availability Statement:** For the samples from the Huntsman Cancer Institute, raw FASTQ files are available through dbGaP (accession phs002262. v1.p1) and processed gene count tables are

## Abstract

Single-cell RNA-sequencing (scRNA-seq) has made it possible to profile gene expression in tissues at high resolution. An important preprocessing step prior to performing downstream analyses is to identify and remove cells with poor or degraded sample quality using quality control (QC) metrics. Two widely used QC metrics to identify a 'low-quality' cell are (i) if the cell includes a high proportion of reads that map to mitochondrial DNA (mtDNA) encoded genes and (ii) if a small number of genes are detected. Current best practices use these QC metrics independently with either arbitrary, uniform thresholds (e.g. 5%) or biological context-dependent (e.g. species) thresholds, and fail to jointly model these metrics in a data-driven manner. Current practices are often overly stringent and especially untenable on certain types of tissues, such as archived tumor tissues, or tissues associated with mitochondrial function, such as kidney tissue [1]. We propose a data-driven QC metric (miQC) that jointly models both the proportion of reads mapping to mtDNA genes and the number of detected genes with mixture models in a probabilistic framework to predict the low-quality cells in a given dataset. We demonstrate how our QC metric easily adapts to different types of single-cell datasets to remove low-quality cells while preserving high-quality cells that can be used for downstream analyses. Our software package is available at https://bioconductor.org/packages/miQC.

## Author summary

We developed the miQC package to predict the low-quality cells in a given scRNA-seq dataset by jointly modeling both the proportion of reads mapping to mitochondrial DNA (mtDNA) genes and the number of detected genes using mixture models in a probabilistic

available through GEO (accession GSE158937). Genome data for the University of Helsinki samples has been deposited at the European Genome-phenome Archive (EGA) which is hosted at the EBI and the CRG, under accession number EGAS00001005066. https://ega-archive.org/studies/EGAS00001005066 The samples were taken as a part of a larger study cohort, where all patients participating in the study provided written informed consent. The study and the use of all clinical material have been approved by The Ethics Committee of the Hospital District of Southwest Finland (ETMK) under decision number EMTK: 145/1801/2015. Wrapper to use miQC with the Seurat analysis package is available at https://github.com/satijalab/seurat-wrappers. All other relevant data are within the manuscript.

**Funding:** AAH, LMW, JAD, CSG, and SCH were supported by the the National Cancer Institute (NCI) from the National Institutes of Health (NIH) grant R01CA237170. SCH was also supported by the National Human Genome Research Institute from the NIH grant R00HG009007 and Alex's Lemonade Stand Foundation. MMF, EPE, KZ, AV were supported by the European Union's Horizon 2020 research and innovation program under Grant Agreement No. 667403 for HERCULES (Comprehensive Characterization and Effective Combinatorial Targeting of High-Grade Serous Ovarian Cancer via Single-Cell Analysis), the Academy of Finland (Projects No.289059, 319243 and 294023), the Sigrid Jusélius Foundation, and the Cancer Foundation Finland. The funders had no role in study design, data collection and analysis, decision to publish, or preparation of the manuscript.

**Competing interests:** The authors have declared that no competing interests exist.

framework. We demonstrate how our QC metric easily adapts to different types of single-cell datasets to remove low-quality cells while preserving high-quality cells that can be used for downstream analyses.

This is a *PLOS Computational Biology* Methods paper.

## Introduction

Recent advances in single-cell RNA-sequencing (scRNA-seq) technologies have enabled genome-wide profiling in thousands to millions of individual cells [2]. As these technologies are relatively costly, researchers are eager to maximize the information gain and subsequent statistical power from each sample and experiment [3]. scRNA-seq is also extremely sensitive to poor or degraded sample quality, which is a particular concern for tissues, such as tumors, obtained during surgeries or other long-duration procedures [4, 5]. It is crucial that 'compromised' cells ('low-quality' or 'failed' cell libraries in the library preparation process or cells that were dead at the time of tissue extraction) be removed prior to downstream analyses to mitigate against discovered results stemming from a technical artifact instead of meaningful biological variation [6, 7]. These considerations have inspired a wealth of research into best practices for quality control (QC) in scRNA-seq [8–10].

One QC metric widely used to identify a compromised cell is if the cell includes a high proportion of sequencing reads or unique molecular identifier (UMI) counts that map to mitochondrial DNA (mtDNA) encoded genes. Mitochondria are heavily involved in cellular stress response and mediation of cell death [11]. These sorts of cellular stresses can be products of the vigorous process of cell dissociation, and the inclusion of these transcriptionally-altered cells can affect downstream analysis outcomes [6]. Additionally, a high abundance of counts mapping to mtDNA genes can indicate that the cell membrane has been broken, and thus cytoplasmic RNA levels are depleted relative to the mRNA protected by the mitochondrial membrane [12]. For these reasons, it is standard practice to remove cells with a large percentage of reads or UMI counts mapping to mtDNA genes using some arbitrary and uniform thresholds, for example greater than 5% [13, 14]. However, recent work has shown these thresholds can be highly dependent on the organism or tissue, the type of scRNA-seq technology used, or the protocol specific decisions made as part of the disassociation, library preparation, and sequencing steps [10, 15, 16]. For instance, evidence suggests that cells that have been treated with certain RNA-preserving reagents prior to library preparation have a much higher mitochondrial fraction compared to fresh tissues [17]. In addition, certain tissues such as kidney tissue can have a high mitochondrial metabolic activity, which can require permissive inclusion thresholds for cells with up to 80% mitochondrial fraction [1].

Other widely used QC metrics to identify compromised cells are the total number of sequencing reads or UMI counts in a sample and the number of unique genes that those reads or counts map to, also known as library complexity [18, 19]. For example, if all observed UMI counts in a cell map to only a few genes, this also suggests that the mRNA in the cell may have been degraded or lost in one or more protocol steps. A standard approach to filter out these cells is to filter out cells with an *ad hoc* threshold of less than a certain number of unique genes detected, such as less than 100 genes. An alternative approach is to rank the cells by their total

UMI count and visually inspect for a knee point in the data, but this approach is often arbitrary and difficult to reproduce [20].

Current best practices filter on these QC metrics independently and often use uniform thresholds, sometimes species-dependent thresholds, which can lead to arbitrary linear classification boundaries (or cutoffs) that may not be appropriate for a given dataset. These cutoffs are often conservative, leaving only a small number of the remaining cells, which offers a non-representative sample of the tissue and constrains downstream analyses. Moreover, filtering on these metrics independently can fail to properly interpret legitimate biological reasons for a low score on a single QC metric, such as quiescent cell populations which will naturally have fewer unique genes expressed, or cells involved in respiratory processes which will naturally have higher mitochondrial expression [14].

Here, we propose an alternative approach to enable researchers to make data-driven decisions about which population a given cell comes from with respect to both mtDNA fraction and library complexity, which is adaptive across scRNA-seq datasets. We aim to create a data-driven decision boundary that can be more inclusive to cell libraries with high library complexity and higher proportion of reads mapping to mtDNA genes. Throughout the rest of the text, we use the terms (i) a *compromised* cell to refer to a low-quality cell that is expected to have few unique genes represented and a high mitochondrial fraction and (ii) an *intact* cell to refer to cells that are of a high-quality (e.g. with an intact cell membrane) with a low proportion of mitochondrial reads and should be included in downstream analyses [12, 21]. For a given scRNA-seq sample, we model the cells using a latent variable model with a latent factor representing the degree of how compromised a given cell is. We fit a finite mixture of models in a probabilistic framework and remove cells based on the posterior probability of coming from the compromised cell distribution. By modeling distributions of parameters for each tissue sample, biological and technical variation can be accounted for in a highly adaptive, sample-specific manner, while still providing a consistent set of principles for inclusion. We demonstrate that across a variety of tissues and experiments, our method preserves more intact cells post-QC than uniform mitochondrial thresholds, which can be used in downstream analyses. Our data-driven methodology for QC is available in a R/Bioconductor software package at https://bioconductor.org/packages/miQC.

## Results

### miQC: A data-driven metric for quality control in scRNA-seq data

To motivate the need of a data-driven approach, we first explored the use of commonly used QC thresholds to remove compromised cells in a high-grade serous ovarian cancer (HGSOC) tissue sample (sample ID 16030X4 from [22] and described in Datasets Section in Methods). For each cell, we calculated the percent of counts mapping to mtDNA genes and the number of unique genes to which those counts map. As stated above, based on previous biological knowledge, we expect intact cells to have low percent of counts mapping to mtDNA genes and moderate to high library complexity. In contrast, compromised cells are expected to have a large percent of counts mapping to mtDNA genes and a low library complexity. In our cancer sample, we observed a peak of counts mapping to mtDNA genes at 13% and a wide range of library complexity values (Fig 1A). However, as the percent of counts mapping to mtDNA genes increases, the number of unique genes decreases significantly, suggesting these are compromised cells. These are the two populations of cells we aim to discover in a data-driven manner.

Using this cancer sample, when we remove cells using a uniform and *ad hoc* QC threshold, for example greater than 10% cell counts mapping to mtDNA genes as suggested by [16], we

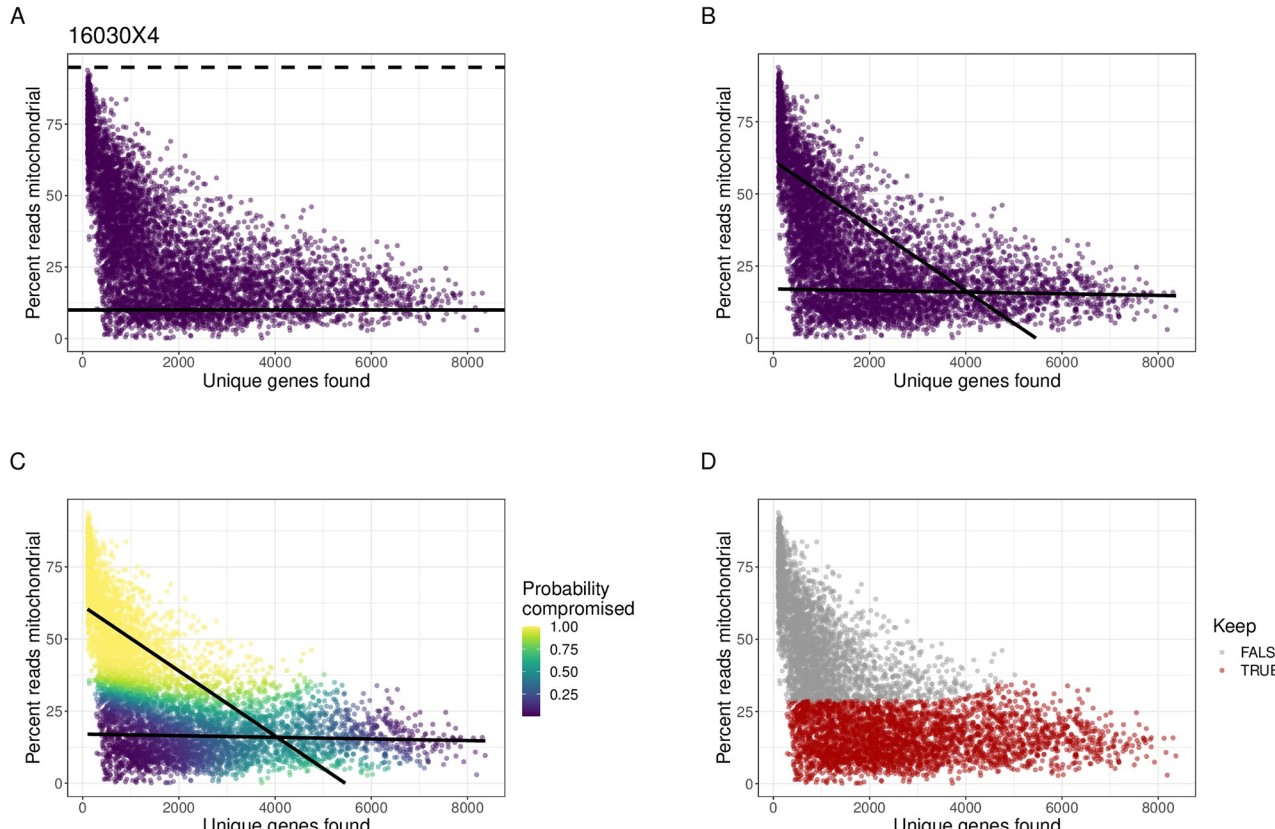

**Fig 1. Uniform and data-driven quality control (QC) thresholds for scRNA-seq data.** Cells ($N = 6618$) from one high-grade serous ovarian cancer (HGSOC) tissue sample (Sample ID: 16030X4) with the number of unique genes detected (x-axis) and percent of cell counts mapping to mitochondrial (mtDNA) genes (y-axis). **(A)** Illustration of removing cells with a uniform QC threshold of greater than 10% cell counts mapping to mtDNA genes (solid black line) and a more data-driven threshold of greater than 3 median absolute deviations (MADs) of the percent of counts mapping to mtDNA genes (dotted black line). **(B)** Using our data-driven approach (miQC), we fit a finite mixture of standard linear regression models with two lines (black lines) to calculate a posterior probability of being a compromised cell. **(C)** Cells shaded by their posterior probability of being compromised. **(D)** Discarding all cells with $\geq$ 75% probability of being compromised creates a data-driven QC threshold for scRNA-seq data.

remove 5828 cells (88.1%) from the sample. As ovarian cancer samples presented the most abundant mtDNA copy numbers in a broad pan-cancer comparison across 38 tumor types [23], mitochondrial transcript content is expected to be relatively high also for intact ovarian cancer cells compared to most normal tissues. Thus, the QC based on an arbitrary limit of 10% is overly aggressive and renders most of the data from a sample unusable. Alternatively, if we use a more data-driven approach that only considers the percent of counts mapping to mtDNA genes and removes cells with greater than 3 median absolute deviations (MADs) [24, 25], we remove no cells from the sample, resulting in an overly permissive QC. Both of these approaches fail in this scenario because they are designed for extremely high-quality datasets with only a trivial number of compromised cells. These analyses motivated our proposed approach that is designed to discriminate between compromised and intact cells that is adaptive to a spectrum of data quality in scRNA-seq data, described in the next section.

**Probabilistic classifications for scRNA-seq data quality using mixtures of linear models.** Because of the limitations of uniform and *ad hoc* QC thresholds, we aimed to use a probabilistic framework that jointly models two QC metrics to predict the compromised cells in a given dataset. We assume that for any cell $i$, there is a latent variable that we do not observe $Z_i = 1$ if the cell is considered a compromised cell and should be removed from downstream

analyses, and $Z_i = 0$ if the cell library is intact. We denote $\pi_1$ as the probability $\Pr(Z_i = 1)$ and $\pi_0 = 1 - \pi_1 = \Pr(Z_i = 0)$. We also define $Y_i$ as the percent of counts in the $i^{th}$ cell that map to mtDNA genes and $X_i$ is the number of unique genes detected or found for the $i^{th}$ cell. Then, we assume that conditional on $Z_i$ and $X_i$, the expected percent of mtDNA counts $Y_i$ is

$$E[Y_i|Z_i = z, X_i = x_i] = f_z(x_i) \tag{1}$$

We note that $f_z$ represents a different function estimated for the two states $z = \{0, 1\}$ (intact or compromised cells, respectively). We assume the errors are modeled with different variance components $\varepsilon_{iz} \sim N(0, \sigma_z^2)$ for the two $z$ states. By default, we assume the function $f_z(x_i)$ takes the form of a standard linear regression model $f_z(x_i) = \beta_{0z} + \beta_{1z}x_i$ where $\beta_{0z}$ represents the mean level of percent of mtDNA counts for the two states $z = \{0, 1\}$ and $\beta_{1z}$ represents the corresponding coefficient, which is also estimated differently for each of the two states $z = \{0, 1\}$ (Fig 1B). This finite mixture of linear regression models is also known as latent class regression [26]. However, our approach can also use a more flexible model such as $f_z(x_i) = \mu_z + g_z(x_i)$ where $\mu_z$ is again the mean level and $g_z(x_i)$ is a nonparametric smooth function that can be estimated with, for example a B-spline basis matrix (S1 Fig). Alternately, for an even simpler model based only on mitochondrial percentage, we can also use a one-dimensional gaussian model where $f_z(x_i) = \mu_z$ (or $x_i$ information is not used) (S1 Fig).

To estimate the parameters $\theta = (\pi_z, f_z)$ for the two states $z = \{0, 1\}$, we use an Expectation Maximization (EM) algorithm [27] implemented in the *flexmix* [28] R package. Using the estimated parameters from the EM algorithm, we calculate the posterior probability of a compromised cell as

$$\gamma_{Z_i}(1) = \Pr(Z_i = 1|Y_i, X_i = x_i, \theta) = \frac{\pi_1 N(Y_i|f_1(x_i), \sigma_1^2)}{\pi_1 N(Y_i|f_1(x_i), \sigma_1^2) + \pi_0 N(Y_i|f_0(x_i), \sigma_0^2)} \tag{2}$$

where $N(\cdot)$ represents the probability density function of a Gaussian distribution with mean $f_z(x_i)$ and variance $\sigma_z^2$ for the two states $z = \{0, 1\}$.

We use the posterior probability $\gamma_{Z_i}(1)$ as the data-driven threshold to exclude (or keep) cells (Fig 1C). In our analyses, we remove cells with a greater than 75% probability of belonging to the compromised cell distribution, in order to maximize the number of potentially informative cells while still removing the cells most likely to confound downstream analyses (Fig 1D). In the next section, we demonstrate how this threshold is adaptive across species, tissues and experimental protocols. However, the `posterior` argument in the miQC package can be used to adjust the posterior probability threshold, depending on the needs of a given experiment.

We also perform two additional processing steps to account for potentially undesired behavior in the posterior distributions. Occasionally, when a cell is far enough below the intact distribution line and near the *x*-intercept of the compromised distribution line, the posterior probability of being compromised is high (S2(A) Fig), causing it to be excluded despite its low percentage of mitochondrial reads (S2(B) Fig). We therefore add a condition that any cell below the intact distribution line is automatically included (S2(C) Fig). Also, some samples have a U-shape in the boundary between kept and discarded cells. When this occurs, cells at the bottom of the trough will be discarded, but some cells with less library complexity (farther left) and higher percentage of mitochondrial reads (higher)—meaning they are worse in both QC metrics being used—will be kept. In samples with a visible U-shape, we identify the cell marked for removal with the lowest mitochondrial percentage, determine its library complexity, and discard all cells with both lower complexity and higher mitochondrial percentage (S2 (D) Fig). This creates a de facto mitochondrial percentage cutoff for all cells with low library

complexity, but is more permissive for cells with high library complexity and high mitochondrial percentage, which are more likely to be intact cells with a biological reason for high mitochondrial expression than their low complexity counterparts [14]. In the miQC package, these corrections are recommended (and the default behavior), but optional using the `keep_all_below_boundary` and `enforce_left_cutoff` parameters.

## miQC is adaptive across species, tissues, and experimental protocols

Previous work has demonstrated that the expected amount of mitochondrial activity varies across species and tissue types. For example, one study concluded that filtering all cells above 5% cell counts mapping to mtDNA genes is appropriate for mouse samples, but that a cutoff of 10% is preferable in human samples [16]. However, it has also been demonstrated that certain tissues, especially those with high energy requirements such as brain, kidney, and heart, have a higher baseline mitochondrial expression [1, 29], and that mtDNA copy numbers highly vary across tissue and cancer types [23].

Here, we consider publicly available scRNA-seq datasets and demonstrate that our miQC approach identifies adaptive QC thresholds across species, tissue types, and experimental protocols. Specifically, we explore $N = 6$ datasets (described in detail in the Datasets Section in Methods) ranging from hundreds to tens of thousands of cells from (i) two species (mouse and human), (ii) five non-cancer tissue types (retinal, immune, brain, pancreas, menstrual blood), (iii) one cancer tissue type (HGSOC), and (iv) two experimental protocols (plate-based and droplet-based single cell protocols).

**Using non-cancer tissues.** Using mouse scRNA-seq data from retinal [30, 31] and immune [32] cells measured on the Drop-seq and Smart-seq2 platforms, we found that miQC identifies a similar QC threshold to using the 5% threshold found in a previous study [16] (Fig 2A, 2B and 2C). However, using mouse scRNA-seq data from brain cells measured on the

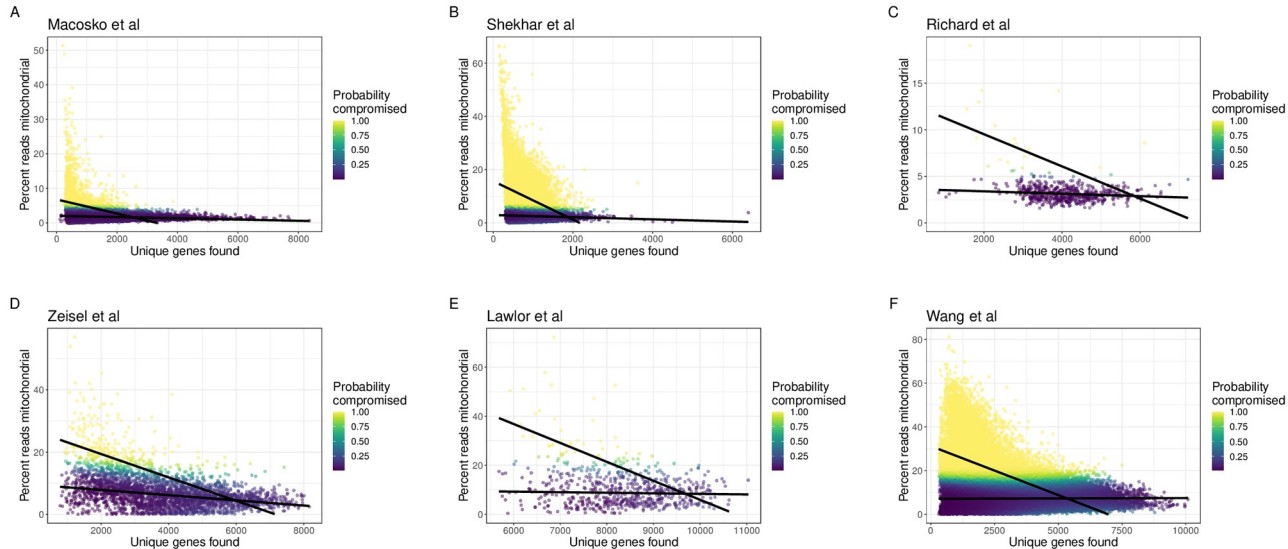

**Fig 2. miQC is adaptive across species, tissues, and experimental protocols.** Using publicly available scRNA-seq data from non-cancer tissues, we calculated the number of detected genes (*x*-axis) and percent of cell counts mapping to mitochondrial (mtDNA) genes (*y*-axis) for each cell. Black lines represent the finite mixture of linear regression models estimated from miQC to adaptively identify compromised cells across datasets. Color represents the posterior probability of a given cell being a compromised cell (yellow is a high probability and purple is a low probability). The data span both **(A-D)** mouse and **(E-F)** human scRNA-seq cells from non-cancer **(A-B)** retinal [30, 31], **(C)** immune [32], **(D)** brain [33], **(E)** pancreas [34], and **(F)** menstrual blood tissues [35]. In addition, the data span **(A, B, F)** droplet-based, **(C)** plate-based, and **(D, E)** microfluidic-based protocols.

Fluidigm C1 platform [33], we found that miQC proposes a less stringent QC threshold compared to the 5% threshold suggested by [16] (Fig 2D). In this case, if a 5% threshold was used, there would be $N$ = 1948 cells (or 64.8%) removed from the sample, which are likely to contain intact and biologically informative cells.

In humans, previous work has shown pancreas typically expresses a large fraction of mtDNA genes [16]. Using human scRNA-seq data from pancreas measured on the Fluidigm C1 platform [34], our miQC approach agrees with this result and the model suggests excluding $N$ = 50 cells (or 7.8%) from the sample (Fig 2E) in contrast to removing $N$ = 290 cells (or 45.5%) if using a 10% threshold as suggested for human tissues. In addition, we found using human scRNA-seq data from menstrual blood measured on the 10x Chromium platform [35] that our miQC approach excludes fewer cells ($N$ = 13499 or 19.0%) from the sample (Fig 2F) in contrast to removing $N$ = 29129 cells (or 41%) if using a 10% threshold as suggested by [16].

**Using cancer tissues.** A major advantage of our data-driven miQC approach is the use of the posterior probability threshold for inclusion, because it allows for a consistent QC metric to be applied across all samples in a set of experiments, while still flexibly accommodating differences in samples or tissues. This is important for experiments leveraging data collected from across different experimental laboratory settings or at multiple times where these factors have been shown to contribute differences in batch effects [36] or percent of counts mapping to mtDNA genes. This is particularly true in application of scRNA-seq cancer samples, where the high heterogeneity of tumor composition and cancer cell behavior make it especially challenging to assign one cutoff metric for all samples.

Here, we apply miQC to a set of scRNA-seq data derived from multiple human high-grade serous ovarian tumors (HGSOC) [22] ($N$ = 7 tumor samples described in detail in the Datasets Section in Methods) using the 10X Chromium experimental protocol. For each cell in a HGSOC tumor sample, we calculate the number of detected genes and the percent of cell counts mapping to mitochondrial genes, similar to Fig 2, which resulted in wide variation of what might be a compromised cell. However, we found that our approach miQC is able to adaptively find QC thresholds across different tumor samples all within the same cancer type (Fig 3). Specifically, we found miQC removes $N$ = 1464 (29.7%), 813 (49.0%), 218 (28.5%), 47 (8.1%), 160 (10.6%), and 448 (12.0%) cells as opposed to 4683 (94.8%), 911 (55.0%), 376 (46.8%), 159 (27.4%), 250 (16.6%), and 324 (8.6%) cells if using a 10% threshold, as is suggested in [16].

## miQC is adaptive across choice of reference genome used in a data analysis

In addition to the biological factors that can affect baseline mitochondrial expression, there are additional technological and experimental factors that can change the observed number of counts mapping to mtDNA genes as well. For example, we found one crucial component is the choice of the reference genome used for quantification of cell reads or UMI counts. The mitochondrial genome has been annotated for decades and genic content is known to be highly conserved across animal species: 37 genes, coding for 13 mRNAs, 2 rRNAs, and 22 tRNAs [37]. However, some reference genomes include all 37 genes where others only include the 13 protein-coding genes.

We investigated this technological confounding factor within one of the HGSOC tumor samples (Sample ID: EOC871). We considered the scRNA-seq cell counts that were quantified using (i) Cell Ranger [2] with the human genome reference GrCh38 (version 2020-A) filtered to remove pseudogenes, and (ii) salmon alevin [38] with the unfiltered human genome reference GENCODE (Release 31) [39]. We found that when quantifying reads with these two different reference genomes, the cell counts that would have mapped to the "missing"

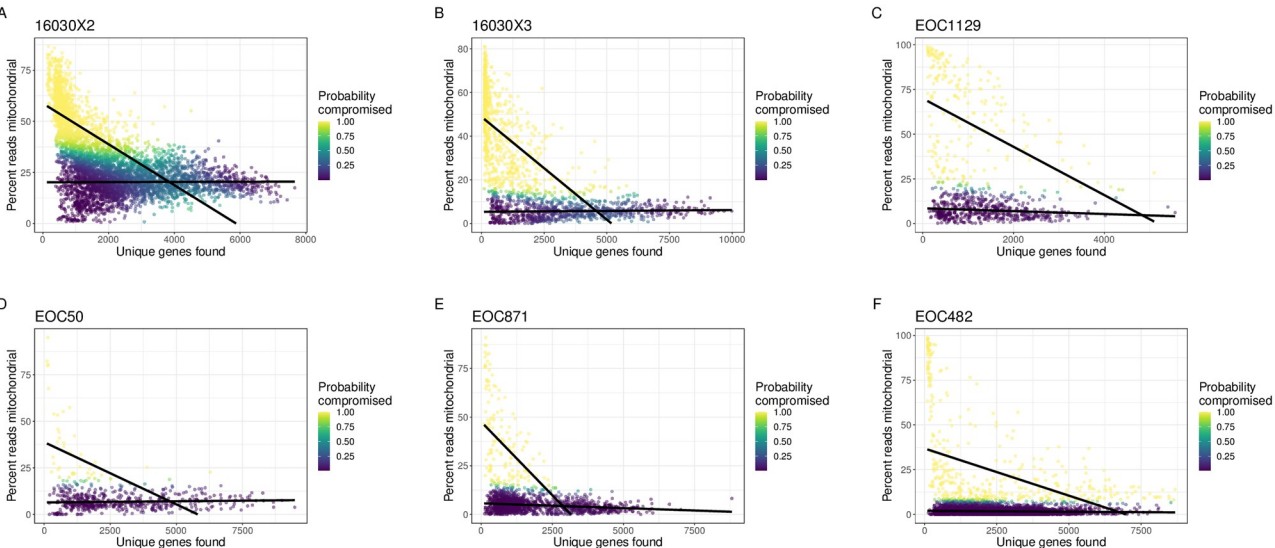

**Fig 3. miQC is adaptive across different samples within same cancer type.** scRNA-seq data from human high grade serous ovarian cancer (HGSOC) samples from **(A-B)** the Huntsman Cancer Institute (Sample IDs: 16030X2, 16030X3) and **(C-F)** the University of Helsinki (Sample IDs: EOC1129, EOC50, EOC871, EOC482). (Data from remaining Huntsman Cancer Institute sample, ID 16030X4, is depicted in Fig 1). We calculated the number of detected genes (*x*-axis) and percent of cell counts mapping to mitochondrial (mtDNA) genes (*y*-axis) for each cell. Black lines represent the finite mixture of linear regression models estimated from miQC to adaptively identify compromised cells. Color represents the posterior probability of a given cell being a compromised cell (yellow is a high probability and purple is a low probability).

mitochondrial tRNA and rRNA genes (in the GENCODE reference genome) are instead assigned to mitochondrial-like pseudogenes on the chromosomes (in the GrCh38 reference genome). This results in a non-uniform shift and technological inflation in the percent of cell counts mapping to mitochondrial genes (Fig 4). These results agree with the findings of Brüning et al that using a filtered transcriptome annotation causes an increase in number of reads mapping to mitochondrial genes irrespective of quantification software used [40]. While we compared GrCh38 and GENCODE annotations, the authors of [40] compared Ensembl annotations with and without cellranger's *mkgtf* function applied, indicating the effect on mitochondrial reads is present across several references. It is important to account for this potential confounding factor if, for example, researchers are performing quality control on cell counts derived with differently derived reference genomes, which we anticipate to become more relevant as cancer atlases grow. Also, mitochondrial reference genomes may diverge further as additional non-coding RNAs and pseudogenes are discovered and characterized [41].

Using a uniform 10% QC threshold to identify compromised cells (Fig 4A and 4B), we found this removes either $N = 101$ and $N = 230$ (or 6.8% and 14.6%) using Cell Ranger and salmon alevin, respectively, when using two different reference genomes, despite these being the exact same cell libraries, just being quantified with two different reference genomes. Interestingly, we also found differences in which cells are removed depending on the choice of reference genome with a greater fraction removed by Cell Ranger (Fig 4C). In contrast, we found our miQC approach (Fig 4D and 4E) is able to flexibly identify different QC thresholds when using two different reference genomes, removing a more similar set of cells: $N = 160$ cells and $N = 136$ cells (or 10.6% and 8.8%) using Cell Ranger and salmon alevin, respectively (Fig 4F). This demonstrates our data-driven approach is able to adjust for differences in this technological confounding factor of diverging mitochondrial annotation in the quantification step of the analysis of scRNA-seq data.

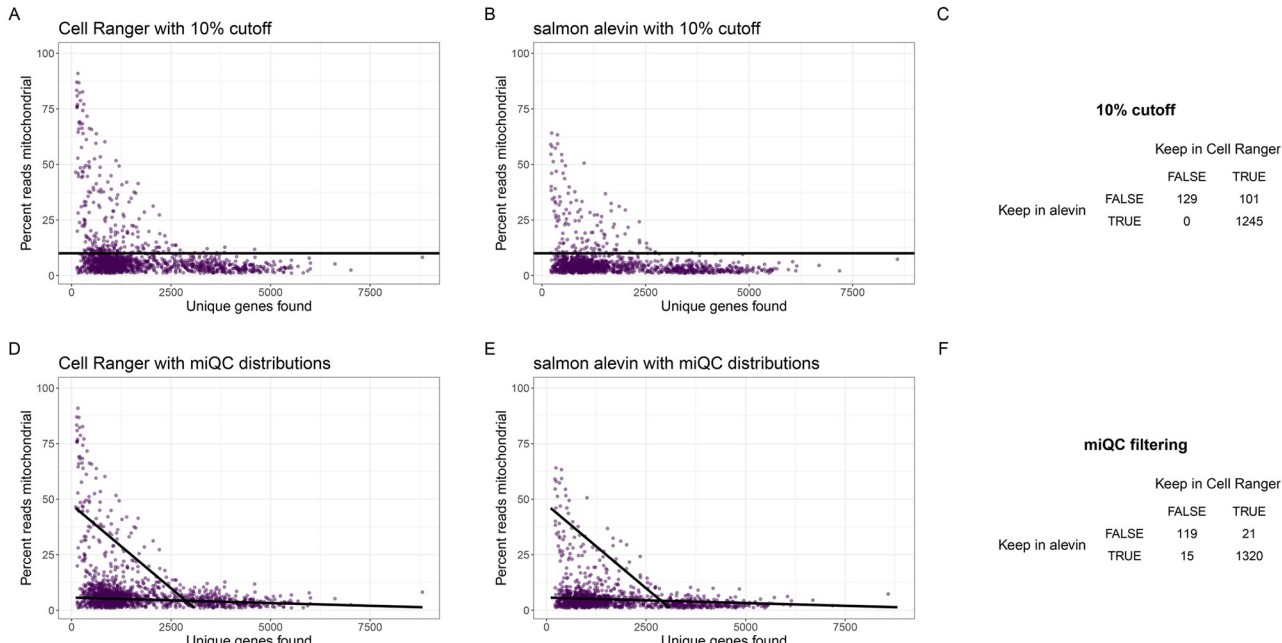

**Fig 4. miQC is adaptive across the choice of reference genome used.** Cells (*N* = 1733) from one high-grade serous ovarian cancer (HGSOC) tissue sample (Sample ID: EOC871) with the number of unique genes found (*x*-axis) and percent of cell counts mapping to mitochondrial (mtDNA) genes (*y*-axis). Quantification of cell counts was performed with **(A)** Cell Ranger using the GrCh38 reference genome (version 2020-A) and **(B)** salmon alevin with the GENCODE reference genome (release 31). In both (A-B), we use a 10% mtDNA threshold (as recommended by Osorio and Cai [16]) for this human cancer sample to remove compromised cells. **(C)** A confusion matrix of how cells are filtered differently by this uniform threshold using the different reference genomes, resulting in a large number of cells removed after QC using either reference genome, but differences in which cells are removed depending on the choice of reference genome. **(D-E)** Using the same tissue sample as (A-C), but here we use our miQC approach to fit a finite mixture of standard linear regression models with two lines (black lines) to calculate the posterior probability of being a compromised cell. **(F)** A confusion matrix of which cells are filtered by our miQC approach which not only results in a larger number of predicted intact cells after QC, but also results in a smaller number of discrepancies between which cells are included after QC.

## miQC minimizes cell type-specific sub-population bias

A standard downstream scRNA-seq data analysis is identifying cell types in a tissue or tumor sample and detecting differences between cell types [25]. A crucial component of this analysis is to have sufficient statistical power to detect differences between cell types, which depends on having appropriate sample sizes of measured cells—and the choice of QC metrics and thresholds directly impacts the number of cells employed in these downstream analyses. For example, in application of unsupervised clustering if a large number of cells are removed post-QC, the number of cells per cluster, and even the number of clusters discovered, can be affected. Therefore, it is important to evaluate whether the choice of QC metric and corresponding threshold do not significantly negatively impact the unsupervised clustering results. In fact, [42] argued "although more stringent filtering tended to be associated with an increase in accuracy, it tended to plateau and could also become deleterious. Most of the benefits could be achieved without very stringent filtering and minimizing subpopulation bias", where *sub-population bias* is defined as disproportionate exclusion of certain cell populations.

Here, we aimed to investigate whether our miQC approach resulted in minimized sub-population bias, as described by [42], compared to the standard recommended approach [16] of using a uniform QC threshold of 10% of cell counts mapping to mtDNA genes. Using one HGSOC tumor sample (Sample ID 16030X4), we preprocessed and normalized the scRNA-seq data according to [25] followed by applying dimensionality reduction using the Uniform

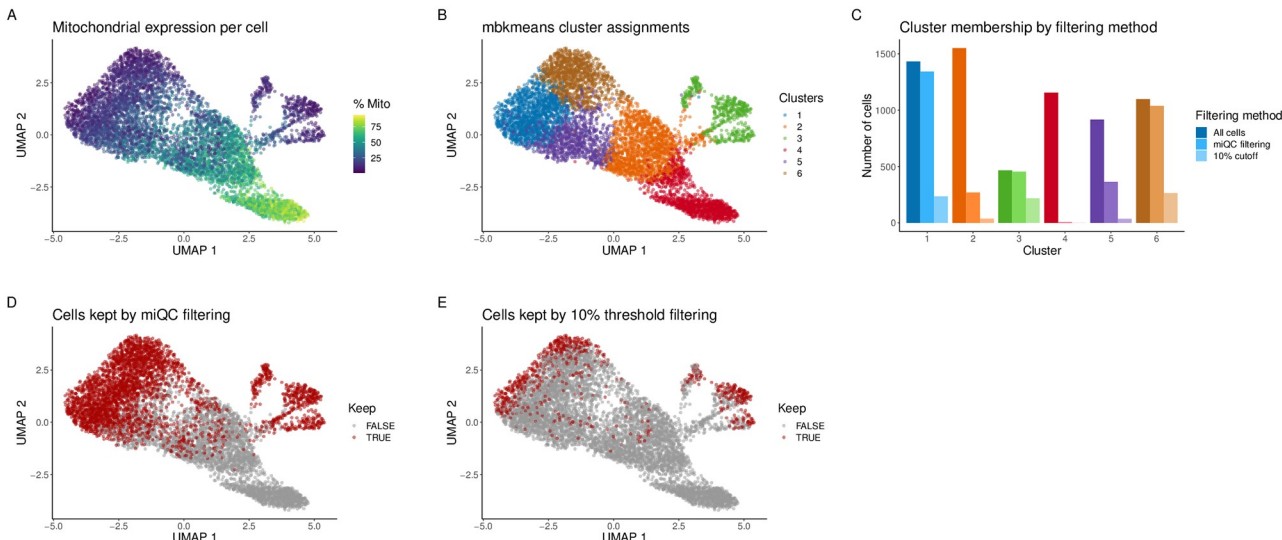

**Fig 5. miQC minimizes sub-population bias in unsupervised clustering.** Cells (*N* = 6691) from one high-grade serous ovarian cancer (HGSOC) tissue sample (Sample ID: 16030X4). **(A)** UMAP representation of cells, colored by percent of cell counts mapping to mtDNA genes. **(B)** Predicted cluster labels for *k* = 6 cell types identified using the mini-batch *k*-means (mbkmeans) algorithm for unsupervised clustering. **(C)** Total number of cells preserved within each predicted cluster group using three filtering approaches: no filtering ('All cells'), miQC filtering, or removing all cells greater than 10% mtDNA content. **(D)** UMAP representation of cells colored by whether miQC keeps (red) or removes (gray) cells post-QC. **(E)** Similar as (D), but using a uniform 10% threshold for mtDNA content.

Manifold Approximation and Projection (UMAP) [43] representation. The percent of cell counts mapping to mtDNA genes in this representation is shown in (Fig 5A). Using the top 50 principal components, we performed unsupervised clustering using the mini-batch *k*-means (mbkmeans) algorithm [44] implemented in the *mbkmeans* [45] R/Bioconductor package for unsupervised clustering to identify cell types, which is a scalable version of the widely-used *k*-means algorithm [46–48] (Fig 5B). The number of clusters (*k* = 6) was determined using an elbow plot with the sum of squared errors (S3 Fig). Using these *k* = 6 clusters, we compared proportions of cells belonging to each predicted cluster using (i) no filtering, (ii) our miQC threshold, and (iii) the uniform QC threshold of 10% of cell counts mapping to mtDNA genes (Fig 5C, 5D and 5E).

We found that the cells in cluster 4 (red bar in Fig 5) were almost entirely removed by miQC and the 10% recommended 10% threshold [16], respectively (cluster 4: 99.6%, 100%). These cells had an average mitochondrial fraction of 68.9% (Fig 5A). We can reasonably infer that those are compromised cells, and as such excluding them from a downstream analysis is appropriate. In contrast, we found that for all other clusters miQC removed far fewer cells than the 10% threshold approach (cluster 1: 6.2%, 83.6%; cluster 2: 82.6%, 97.6%; cluster 3: 2.6%, 53.2%; cluster 5: 60.3%, 96.1%; cluster 6: 5.4%, 75.9%). This suggests miQC preserves more cells within each predicted cluster and minimizes sub-population bias, compared to the uniform threshold approach of a 10% cutoff.

## Methods

### Ethics statement

All patients participating in the study provided written informed consent. An Institutional Review Board approved the protocol at University of Pennsylvania (832353), Johns Hopkins University (00009288), Univeristy of Utah (118086), and University of Colorado (21–2761).

The use of clinical material from University of Helsinki was approved by The Ethics Committee of the Hospital District of Southwest Finland (ETMK) under decision number EMTK: 145/1801/2015.

## Datasets

**Non-cancer tissue scRNA-seq datasets.** We obtained non-cancer tissue scRNA-seq datasets for the studies [30], [31], [32], [33], and [34] from the R/Bioconductor data package *scRNAseq* [49]. We obtained the scRNA-seq data from [35] from the Sequence Read Archive (SRA) (accession code SRP135922). All datasets were processed using *scater* [24], as described in the "Preprocessing scRNA-seq datasets" section. Table 1 contains a summary of the non-cancer datasets used: the organism, the tissue, the experimental protocol, and the number of cells prior to QC for each dataset.

**Cancer tissue scRNA-seq datasets.** The $N = 7$ HGSOC tumor samples were collected and sequenced at Huntsman Cancer Institute, Utah, USA ($N = 3$) and at University of Helsinki, Finland ($N = 4$).

For the samples from the Huntsman Cancer Institute, raw FASTQ files are available through dbGaP (accession phs002262.v1.p1) and processed gene count tables are available through GEO (accession GSE158937) [22]. Complete details of the experimental protocol and sequencing steps followed for these tumor samples are provided in [22], but in brief library prepration was performed using 10x Genomics 3' Gene Expression Library Prep v3, and sequencing was done on an Illumina NovaSeq instrument. Quantification for these samples was performed using salmon alevin [38] with a index genome generated from GENCODE v31 [39].

Genome data for the University of Helsinki samples has been deposited at the European Genome-phenome Archive (EGA) which is hosted at the EBI and the CRG, under accession number EGAS00001005066. The samples were taken as a part of a larger study cohort, where all patients participating in the study provided written informed consent. The study and the use of all clinical material have been approved by The Ethics Committee of the Hospital District of Southwest Finland (ETMK) under decision number EMTK: 145/1801/2015. Immediately after surgery, tissue specimens were incubated overnight in a mixture of collagenase and hyaluronidase to obtain single-cell suspensions. Cell suspensions were passed through a 70-$\mu$m cell strainer to remove cell clusters and debris and centrifuged at 300 x g. Cell pellets were resuspended in a resuspension/washing buffer (1X PBS supplemented with 0.04% BSA) and washed three times. scRNA-seq libraries were prepared with the Chromium Single Cell 3' Reagent Kit v. 2.0 (10x Genomics) and sequenced on an Illumina HiSeq4000 instrument. Using the raw FASTQ files, we performed the quantification step with two different methods to obtain two different UMI counts matrices. First, Cell Ranger (version 3.1.0) [2] was used to perform sample de-multiplexing, alignment, filtering, and barcode and UMI quantification.

**Table 1. Description of non-cancer tissue scRNA-seq datasets.** Columns from left to right include the source, the organism, the type of tissue, the experimental protocol, and the number of cells prior to QC in each dataset.

| Source | Organism | Tissue | Protocol | Cells |
| --- | --- | --- | --- | --- |
| Richard et al. [32] | Mouse | T cells | Smart-seq2 | 572 |
| Zeisel et al. [33] | Mouse | Brain | Fluidigm C1 | 3005 |
| Shekhar et al. [31] | Mouse | Retina | Drop-seq | 44994 |
| Macosko et al. [30] | Mouse | Retina | Drop-seq | 49300 |
| Lawlor et al. [34] | Human | Pancreatic islets | Fluidigm C1 | 638 |
| Wang et al. [35] | Human | Menstrual blood | 10X Chromium | 71032 |

**Table 2. Description of human high-grade serous ovarian cancer (HGSOC) tissue scRNA-seq datasets.** Columns from left to right include the source, the organism, the location from where the tumors were obtained, and the number of cells prior to QC in each dataset. Huntsman Cancer Institute samples were processed with the 10x Genomics 3' Gene Expression Library Prep v3 protocol, and University of Helsinki samples were processed with Chromium Single Cell 3' Gene Expression v2 protocol.

| Source | Sample ID | Location | Cells |
|---|---|---|---|
| Weber et al. (2020) [22] | 16030X2 | Huntsman Cancer Institute | 4939 |
| Weber et al. (2020) [22] | 16030X3 | Huntsman Cancer Institute | 1725 |
| Weber et al. (2020) [22] | 16030X4 | Huntsman Cancer Institute | 6691 |
| Novel as part of manuscript | EOC1129 | University of Helsinki | 1086 |
| Novel as part of manuscript | EOC50 | University of Helsinki | 930 |
| Novel as part of manuscript | EOC871 | University of Helsinki | 1733 |
| Novel as part of manuscript | EOC482 | University of Helsinki | 3879 |

GRCh38.d1.vd1 genome was used as reference and GENCODE v25 for gene annotation. Second, salmon alevin [38] (version 1.4.0) was also used with GRCh38.p13 as reference genome and GENCODE v34 for gene annotation. Table 2 contains a summary of the cancer datasets used, including the source, the organism, the location from where the tumors were obtained, and the number of cells prior to QC in each dataset.

## Data analysis

**Preprocessing scRNA-seq datasets.** We processed the gene-by-cell matrix from each dataset using the *scater* [24] R/Bioconductor package, including calculating the number of unique genes represented and the percent of reads or UMI counts mapping to mtDNA genes. At this step, we removed any cells with fewer than 500 total reads or fewer than 100 unique genes represented, which we considered to be unambiguously failed.

To represent the effect of miQC on downstream analyses, we calculated and plotted the Uniform Manifold Approximation and Projection (UMAP) representation of the single-cell expression data using functions in the *scater* package. We chose to highlight how miQC filtering specifically affects clustering results using the *mbkmeans* package, which uses mini-batches to quickly and scalably produce k-means clustering assignments [45]. We ran mbkmeans on a reduced representation of our expression data, the first 50 principal components as calculated via scater. All visual representations and figures were generated using the *ggplot2* R package [50].

**miQC software implementation.** We used the R package *flexmix* [28] to fit the finite mixture of linear (or non-linear) models, depending on the functional form of $f_z(x_i)$ used. The *flexmix* R packages performs estimation of the parameters using an Expectation-Maximization (EM) algorithm [27]. Like all implementations of the EM algorithm, *flexmix* is not guaranteed to find a global maximum likelihood, meaning that users should check for convergence across multiple initializations. In our case with a finite mixture of two standard linear regression models, we found that *flexmix* converges to extremely similar parameters for each iteration of a given sample, but that the order of distributions given in each iteration is non-deterministic. Therefore, we assumed that the distribution with the greater *y*-intercept, meaning the distribution with higher percent of cell counts mapping to mtDNA genes at low library complexity, was the compromised cell distribution. The parameters estimated from each mixture model were used to calculate the posterior probability of a cell coming from the compromised cell distribution. Our miQC software is available as an R/Bioconductor package under a BSD-3-Clause License at https://bioconductor.org/packages/miQC. The code used to download

datasets, perform the analyses, and reproduce the figures is available at https://github.com/greenelab/mito-filtering.

## Discussion

One critical assumption of our model is that mitochondrial reads are not informative in terms of biological variation. While this is true in many contexts, there are some contexts where high mitochondrial expression is biologically relevant and informative. For instance, scRNA-seq data has shown that aberrant mitochondrial activation is implicated in development of polycystic ovary syndrome (PCOS) [51]. Removing all cells with a large percentage of mitochondrial reads in a PCOS study would therefore hinder much of the downstream analysis. More broadly, metabolic shifts between oxidative phosphorylation and glycolysis, an important indicator of cell proliferation, can also increase or decrease mitochondrial expression [52, 53].

Generally, researchers are able to assess if mitochondrial expression may be relevant to their experimental question at hand. In the majority of cases, cells with a large percentage of mitochondrial reads–especially when paired with few uniquely expressed genes or low numbers of total counts (reads or UMIs)–can be reasonably interpreted as a sign of cell damage and those cells should be discarded.

Our miQC mixture model is designed for scenarios in which there are a non-trivial amount of compromised cells and the amount of compromised cells might vary across samples or experiments. For scRNA-seq data generated from archived tumor tissues, this is often the case. However, in optimal conditions where there are no or few damaged cells, the mixture model may not be able to accurately estimate parameters for the compromised cell distribution, as there might only be a handful of compromised cells. In this case, the model is thus liable to choose very similar parameters for the two distributions, causing the probabilistic assignments for individual cells to be unstable and good cells to be excluded unnecessarily. As an example, our tumor sample EOC50 (Fig 3D) had no cells with an extremely high mitochondrial fraction, meaning the intercept for the "compromised" cell distribution was fitted at a much lower value than the other tumors. In this case, miQC actually excluded more cells than a simple 10% mitochondrial threshold did. With this in mind, for cases with few to no concerns about tissue quality, we recommend using Median Absolute Deviation (MAD) as a data-driven approach for filtering out a small number of damaged cells [54]. We also caution against using miQC on data that has already been filtered by some prior preprocessing step, and recommend users of miQC be aware of any filtering that has been done on their data, especially in the case of public datasets.

It is possible that not only tissue types may have different baselines of mitochondrial expression, but that the baselines would also vary across the cell types within a heterogeneous tissue, such as a dissociated tumor [16]. This suggests an extension of our miQC approach for future development where the intact/compromised distribution parameters could be estimated for each cell type independently. However, in most scRNA-seq experiments involving tissues, cell type identities are not known *a priori* and cannot be determined without first performing QC. Stratifying by cell type is thus currently not advisable for the main uses of miQC.

miQC is a highly flexible model, and as such it is conceptually and practically simple to adjust to a given user's needs. We have designed the miQC package with the most common use-cases built in, with the two most commonly used QC metrics, percent mitochondrial reads and number of unique genes, used to build the model. However, there are many other metrics that can be used for QC, including percent of reads mapping to ribosomal genes, total number of UMIs detected, number of features to which 50% of the reads map, etc. These metrics can

be calculated using *scater* [24], with which miQC is designed to work. These metrics can then be passed to *flexmix* using a similar notation as any linear model in R (see miQC software package for examples). Many of these metrics will be highly correlated with one another, so we caution users to check for multicollinearity before including a large number of variables in their model [55].

## Conclusion

Ensuring the quality of scRNA-seq data is essential for robust and accurate transcriptomic analyses. Percent of reads mapping to the mitochondria is a very useful proxy for cell damage, but existing QC methods do not do justice to the myriad of biological and experimental factors relevant to mitochondrial expression. The standard wisdom of removing all cells with greater than 5% (or 10%) mitochondrial counts is unnecessarily stringent in many tissue types, especially cancer tissues, causing a massive loss of potentially informative cells. Our new method, miQC, offers a probabilistic approach to identifying high-quality cells within an individual sample, based on the assumption that there are both intact and compromised cells within the samples with associated characteristics. This method is flexible and adaptive across experimental platforms, organism and tissue types, and disease states. It is robust to technical differences that alter standard QC metrics, such as differences in reference genome. It also maximizes the information gain from an individual experiment, often preserving hundreds or thousands of potentially informative cells that would be thrown out by uniform QC approaches. miQC is now available as a user-friendly R package available at https://bioconductor.org/packages/miQC, allowing researchers to tailor their QC to the needs of a given scRNA-seq dataset and experiment in a consistent way.

## Supporting information

**S1 Fig. miQC is extensible to a combination of linear and non-linear models. (A)** A mixture model on high-grade serous ovarian tumor data, where the intact cell distribution is modeled linearly and the compromised cell distribution is modeled using a b-spline.**(B)** Posterior probability of tumor cells belonging to compromised distribution as fitted with a spline model. **(C)** Cells with greater than 75% posterior probability of being compromised are marked for removal, after the two default corrections (keep_all_below_boundary = TRUE and enforce_left_cutoff = TRUE). **(D-F)** The same tumor, but with a one-dimensional Gaussian mixture model on percent mitochondrial reads.
(TIF)

**S2 Fig. Conditional corrections for linear models. (A)** The posterior distribution of linear mixture models from Fig 1, shown again for reference. **(B)** Filtering based only using a 66% posterior threshold with no default corrections (keep_all_below_boundary = FALSE and enforce_left_cutoff = FALSE) Low-mitochondrial cells slated for removal are circled in black. **(C)** Same as (B), but including the keep_all_below_boundary = TRUE parameter, where all cells with lower mitochondrial fraction than the predicted intact distribution are kept. This prevents the group of cells in the bottom center from being excluded. **(D)** Same as (B), but including both default corrections (keep_all_below_boundary = TRUE and enforce_left_cutoff = TRUE). This corrects the U-shape boundary. The excluded cell with the lowest mitochondrial percentage is identified, and any cell with both greater mitochondrial percentage and lower library complexity is excluded.
(TIF)

**S3 Fig. Selecting an appropriate number of clusters for tumor data.** We ran mbkmeans on our tumor data for a range of $k$ from 2 to 10. Based on the within cluster sum of squares (WCSS), we proceeded with 6 clusters.
(TIF)

## Acknowledgments

We thank John Wherry for consultation on ovarian cancer biology, as well as members of Greene Lab for scientific feedback, particularly Alexandra Lee for code review. We thank the High-Throughput Genomics Shared Resource at the Huntsman Cancer Institute at University of Utah for assistance with data generation. We would also like to thank Johanna Hynninen (Turku University Hospital), as well as Katja Kaipio, Kaisa Huhtinen, Tarja Lamminen and Naziha Mansuri (University of Turku) for the surgery and pre-processing of University of Helsinki HGSOC samples, respectively. We thank Diego Espinoza for writing a wrapper to use miQC with the Seurat analysis package, which is available at https://github.com/satijalab/seurat-wrappers.

## Author Contributions

**Conceptualization:** Stephanie C. Hicks.

**Data curation:** Ariel A. Hippen.

**Formal analysis:** Ariel A. Hippen.

**Funding acquisition:** Jennifer Anne Doherty, Anna Vähärautio, Casey S. Greene, Stephanie C. Hicks.

**Investigation:** Ariel A. Hippen, Matias M. Falco, Lukas M. Weber, Erdogan Pekcan Erkan, Kaiyang Zhang.

**Methodology:** Ariel A. Hippen, Stephanie C. Hicks.

**Resources:** Matias M. Falco, Erdogan Pekcan Erkan, Jennifer Anne Doherty, Casey S. Greene, Stephanie C. Hicks.

**Software:** Ariel A. Hippen, Matias M. Falco, Lukas M. Weber.

**Supervision:** Anna Vähärautio, Casey S. Greene, Stephanie C. Hicks.

**Visualization:** Ariel A. Hippen.

**Writing – original draft:** Ariel A. Hippen, Stephanie C. Hicks.

**Writing – review & editing:** Ariel A. Hippen, Matias M. Falco, Lukas M. Weber, Erdogan Pekcan Erkan, Jennifer Anne Doherty, Anna Vähärautio, Casey S. Greene, Stephanie C. Hicks.

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
