## [Decision Letter · Decision Letter 0]

26 Apr 2021

Dear Dr. Hicks,

Thank you very much for submitting your manuscript "miQC: An adaptive probabilistic framework for quality control of single-cell RNA-sequencing data" for consideration at PLOS Computational Biology.

As with all papers reviewed by the journal, your manuscript was reviewed by members of the editorial board and by several independent reviewers. In light of the reviews (below this email), we would like to invite the resubmission of a significantly-revised version that takes into account the reviewers' comments.

We cannot make any decision about publication until we have seen the revised manuscript and your response to the reviewers' comments. Your revised manuscript is also likely to be sent to reviewers for further evaluation.

Sincerely,

Magnus Rattray

Guest Editor

PLOS Computational Biology

Ilya Ioshikhes

Deputy Editor

PLOS Computational Biology

Reviewer's Responses to Questions

**Comments to the Authors:**

Reviewer #1: My main review comments are in the "miQC-review.pdf" file, generated from "review.Rmd". Other PDFs are supporting material to illustrate points made in that file. The Makefile defines rules to create each other PDF by rendering "comparison.Rmd" on different datasets.

I have included the code used in case it helps, and also in case I have made some error(s) that affect the comments made in my review.

Reviewer #2: The authors propose a probabilistic mixture model called miQC that simultaneously models the proportion of reads mapping to mitochondrial genes (mtDNA) and the number of detected genes (nUMI).

As the authors point out, the current "state of the art" is to set arbitrary thresholds on the distributon of mtDNA and nUMIs and there is a need for a multi-dimensional probabilistic model for QC of scRNA-seq data. The use of posterior probabilities to predict low-quality cells that miQC provides is handy, but the method is overall too simplistic and does not convincingly show improvement over current approaches. I think miQC is a useful package to complement scater/scran, but I do not think it delivers a sufficient advance to warrant publication as an article in PLOS Computational Biology. I am sorry that I cannot provide a more positive response and I hope my comments below are useful to the authors:

Major comments:

- The authors state that "Current best practices use all these QC metrics independently and commonly use uniform thresholds". I do not think this is true. Just looking at the vignettes of (arguably) the two most commonly used packages for quality control of scRNA-seq (Seurat and scran/scater), both of them include scatterplots of mtDNA and nUMI to select low quality cells (https://bioconductor.org/packages/release/bioc/vignettes/scater/inst/doc/overview.html, https://satijalab.org/seurat/articles/pbmc3k_tutorial.html). As I mentioned above, a posterior probability is preferable over setting a threshold manually, but in practice, if using only two QC dimensions these two approaches are almost equivalent (as shown in Fig1D, where one could simply set a threshold of ~30 mtDNA content).

- mtDNA content and nUMI are the two QC variables most commonly used in practice and the authors were right to choose them as input for their model. However, if one aims to use a latent variable model to define low quality cells I would include additional measurements in the model such as the fraction of features account for ~50% of the reads, the ribosomal fraction, etc. Most of these variables are highly correlated, but a latent variable model approach should be able to exploit such covariation patterns to provide better probabilistic decisions for low-quality cells.

- In Fig5 the authors compared miQC to the "the standard (?) approach of using a uniform QC threshold of 10% of cell counts mapping to mDNA genes". I do not think this is a fair comparison. The mtDNA QC threshold value is not universal and varies from experiment to experiment (even between samples from the same tissue).

- In practice, the QC thresholds are often adjusted after inspection of the latent manifolds (i.e. the UMAPs). Is there a way to incorporate the latent manifold as additional information for the probabilistic decision of low-quality cells? The package could automatically generate multiple UMAP plots under different QC thresholds and then try to come with an optimal threshold that maximises the number of intact cells but at the same time excludes clusters with compromised cells. If this approach is implemented efficiently, I think this could make a significant improvement in the current (tedious) QC pipelines

Reviewer #3: This manuscript a new method and R software package to improve the quality control (specifically filtering out) of "compromised" (problematic or low quality) cells from scRNA-seq datasets prior to further downstream analysis.

The idea itself is simple: use the two most important and most commonly used metrics for cell QC in scRNA-seq analyses (number of genes with non-zero observed expression and percentage of expression from mitochondrial genes), but instead of applying hard thresholds as most people do, instead use a mixture model to identify populations of "compromised" and "good" cells, with a posterior probability of a cell being "compromised". The posterior probability can then be used as a threshold (the authors propose filtering out cells with a posterior probability of being compromised >75%) that combines the information about cell quality from the two underlying metrics.

I view this paper as a relatively simple idea well executed. The method is simple enough to be transparent to the user (an advantage) and yet it does advance current (typical) practices in cell QC for scRNA-seq data. The authors convincingly demonstrate miQC's advantages over hard thresholding in several settings. They also offer a clear discussion of when miQC is not the best option, which is important information and appreciated.

As a user I found the software package easy to install and to use following the vignette provided with the package. The vignette is well written - clear and easy to follow. I was easily able to work through the code examples provided and it gave me the information I would need to use miQC in my own data analyses. The

method is very fast on a dataset of 3005 cells, and I have no concerns about it scaling reasonably to larger datasets. Credit to the authors for making their open-source code for the package and also code to reproduce the work presented in the manuscript easily available.

The manuscript itself is clear, well-structured and well-written throughout. I have only a few minor comments and happily recommend publication of this work. I note that the authors plan to release the software package through Bioconductor, which I heartily support - this method will be a useful and welcome addition to

the Bioconductor ecosystem for scRNA-seq data analysis.

Minor:

- Last sentence of abstract: "package is available in at https://..." -> delete "in"

- Suggest "extensive" on l7 is unnecessary

- l54: "using latent variable" -> "using a latent variable"

- l214: "This highlights..." - this what? Add a noun

- l223: "being being"

- l279: "Section ." - actual section label/number is missing in this sentence

- l336: "mixture two" -> "mixture of two"

- l340: "meaning the for the cells with a low library complexity" ?? clarify

- l340-41: "we labeled distribution" -> "we labeled the distribution"

- l342: "was" -> "were" (subject-verb agreement)

- Working through the vignette I did receive the following warning, which is not shown in the vignette:

> plotModel(sce, model) + viridis::scale_fill_viridis()

Warning message:

Removed 54 row(s) containing missing values (geom_path).

The authors will be able to determine if that's expected behaviour that is simply suppressed for clarity in the vignette or an issue that they wish to follow up on.

**Have the authors made all data and (if applicable) computational code underlying the findings in their manuscript fully available?**

Reviewer #1: Yes

Reviewer #2: None

Reviewer #3: Yes

PLOS authors have the option to publish the peer review history of their article (what does this mean?). If published, this will include your full peer review and any attached files.

Reviewer #1: **Yes: **Alan O'Callaghan

Reviewer #2: No

Reviewer #3: **Yes: **Davis J. McCarthy
---

## [Decision Letter · Decision Letter 1]

20 Jul 2021

Dear Dr. Hicks,

We are pleased to inform you that your manuscript 'miQC: An adaptive probabilistic framework for quality control of single-cell RNA-sequencing data' has been provisionally accepted for publication in PLOS Computational Biology.

I would like to add some comments over the decision for you and for the reviewers. The reviewers have not reached a consensus in this case and specifically there was a difference of opinion from the three reviewers over the degree of novel contribution in the work. Two were very supportive overall while another reviewer felt there was not enough methodological novelty, although nevertheless acknowledging the work to be correct and a potentially useful contribution. I have made the decision to go with the majority view since I think that judging the degree of methodological novelty sufficient for publication is a subjective matter. The work looks like a useful and well executed contribution from my perspective. 

Best regards,

Magnus Rattray

Guest Editor

PLOS Computational Biology

Ilya Ioshikhes

Deputy Editor

PLOS Computational Biology

Reviewer's Responses to Questions

**Comments to the Authors:**

Reviewer #1: I thank the authors for a thorough and considered response to all of my comments, and for the revisions made to their method and manuscript. I consider their manuscript as it now stands to be a rigorous and important contribution to the field.

I am happy to recommend this manuscript for publication and look forward to using the authors' method in my own work.

Reviewer #2: The authors have addressed some of my comments, but have not made significant improvements on the methodology. The use of bivariate relationships for QC of scRNA-seq data is useful but not novel. The proposed approach turns the bivariate relationship into posterior probabilities by using a few lines of code that calls FlexMix, an already existing software. The authors have demonstrated that this gives sensible results, but I am not convinced that this delivers a sufficient advance to warrant publication as an article in PLOS Computational Biology.

**Have the authors made all data and (if applicable) computational code underlying the findings in their manuscript fully available?**

Reviewer #1: Yes

Reviewer #2: None

PLOS authors have the option to publish the peer review history of their article (what does this mean?). If published, this will include your full peer review and any attached files.

Reviewer #1: **Yes: **Alan O'Callaghan

Reviewer #2: No

---

## [Editor Report · Acceptance letter]

19 Aug 2021

PCOMPBIOL-D-21-00428R1 

miQC: An adaptive probabilistic framework for quality control of single-cell RNA-sequencing data

Dear Dr Hicks,

I am pleased to inform you that your manuscript has been formally accepted for publication in PLOS Computational Biology. Your manuscript is now with our production department and you will be notified of the publication date in due course.

With kind regards,

Olena Szabo
